# Mapping Flash Flood Hazards in Arid Regions Using CubeSats

Zhaocheng Wang and Enrique R. Vivoni *

School of Sustainable Engineering and the Built Environment, Arizona State University, Tempe, AZ 85287, USA
* Correspondence: vivoni@asu.edu

**Abstract:** Flash flooding affects a growing number of people and causes billions of dollars in losses each year with the impact often falling disproportionally on underdeveloped regions. To inform post-flood mitigation efforts, it is crucial to determine flash flooding extents, especially for extreme events. Unfortunately, flood hazard mapping has often been limited by a lack of observations with both high spatial and temporal resolution. The CubeSat constellation operated by Planet Labs can fill this key gap in Earth observations by providing 3 m near-daily multispectral imagery at the global scale. In this study, we demonstrate the imaging capabilities of CubeSats for mapping flash flood hazards in arid regions. We selected a severe storm on 13–14 August 2021 that swept through the town of Gila Bend, Arizona, causing severe flood damages, two deaths, and the Declaration of a State of Emergency. We found the spatial extent of flooding can be mapped from CubeSat imagery through comparisons of the near-infrared surface reflectance prior to and after the flash flood event ($\Delta$NIR). The unprecedented spatiotemporal resolution of CubeSat imagery allowed the detection of ponded ($\Delta$NIR $\leq -0.05$) and flood-affected ($\Delta$NIR $\geq +0.02$) areas that compared remarkably well with the 100-year flood event extent obtained by an independent hydraulic modeling study. Our findings demonstrate that CubeSat imagery provides valuable spatial details on flood hazards and can support post-flood activities such as damage assessments and emergency relief.

**Keywords:** small satellites; hydrometeorology; urban flooding; drylands; monsoon; near-infrared

## 1. Introduction

Flash floods, recognized as a common and catastrophic natural hazard, have caused significant fatalities and economic damages worldwide. According to the U.S. Natural Hazard Statistics, flash flooding was the second deadliest (135 fatalities) hazard and caused the third most economic damage (USD 2456 million) in the year 2021 [1]. Moreover, flash flood damages are subject to compound effects of climate change (more frequent extreme precipitation events) [2,3], population growth [4], aging infrastructure [5,6], and varying social vulnerability [7,8]. The tremendous societal and economic hazard of flash flooding necessitates the development of more reliable monitoring tools, including forecasts [9], nowcasts [10,11], and real-time warning systems [12] to inform decision-making, as well as post-event analysis systems to determine flooding extent. These efforts can benefit various hazard mitigation measures, including emergency rescue, damage assessment, flood insurance claims, and granting of flood relief funding.

A region particularly prone to flash flood hazards is the arid Southwest U.S. [13,14], where most summer precipitation occurs as high-intensity convective thunderstorms during the North American Monsoon (NAM). Rainfall events in the NAM season contribute to 35–45% of the annual precipitation in the Southwest U.S. [15,16]. In addition, extreme thunderstorms also produce severe, sometimes destructive, flash floods [14,17,18], as well as damaging debris flows from hillslope failures [19,20]. The spatial organization of flood-producing storms is highly variable in space and time due to the interactions between synoptic conditions, mesoscale processes, and terrain [18,21]. In addition, flood responses are complicated by the inherent uncertainties in the rainfall-runoff transformation [22,23], especially in areas experiencing urban growth [24,25].

Despite their importance, the ability to map flash flood impacts is limited due to the current reliance on water-level sensors, which are typically sparse in a region [26]. In comparison, spatially continuous remote sensing observations are effective in mapping the areal extent of flooding, providing valuable information for post-flood damage analysis [27–29]. Flooding effects are often evident in the near-infrared surface reflectance due to the sharp contrast between standing water (or moist soil) and relatively dry surrounding areas [30]. However, most satellite sensors have either low spatial resolution or low revisiting times, limiting their capacity to detect monsoon-induced flooding which can have a short duration (~1 to 12 h) over small spatial extents (~1 to 10 km$^2$) [22,31]. In recent years, CubeSat imagery acquired from a constellation of small satellites has begun to provide an unprecedented, near-daily, global mapping capacity at 3 m resolution [32–34], which is ideal for the mapping of monsoon flood hazards in the Southwest U.S.

In this study, we selected a fatal flash flood event (13–14 August 2021) in central Arizona (town of Gila Bend) to demonstrate the applicability of CubeSat imagery in mapping flooding extents. This study is organized as follows. We first present a brief background on the study area and the flash flood event and then describe the datasets and methods utilized (Section 2). We subsequently introduce the analysis of the storm event, precipitation and streamflow data, and spatial flood mapping from CubeSat imagery (Section 3). In Section 4, we discuss the implications of CubeSat imagery in post-flood analyses for arid regions. Section 5 concludes this study.

## 2. Datasets and Methodology

### 2.1. Study Area

This study focuses on the town of Gila Bend, with a population of roughly 1900, located in the southwestern portion of Maricopa County (~115 km southwest of Phoenix) in central Arizona (Figure 1a). The town has the typically hot and arid climate of the Sonoran Desert, with an average daily temperature of 23 °C and annual precipitation of 179 mm/year based on 1981–2010 climate normals from National Climatic Data Center. Most rainfall arrives during the summer (July–September) and winter (December–February) seasons, accounting for 35% and 54% of the mean annual precipitation, respectively. The dominant land cover types include desert shrublands, agricultural land (much of which is fallowed), and small urbanized areas.

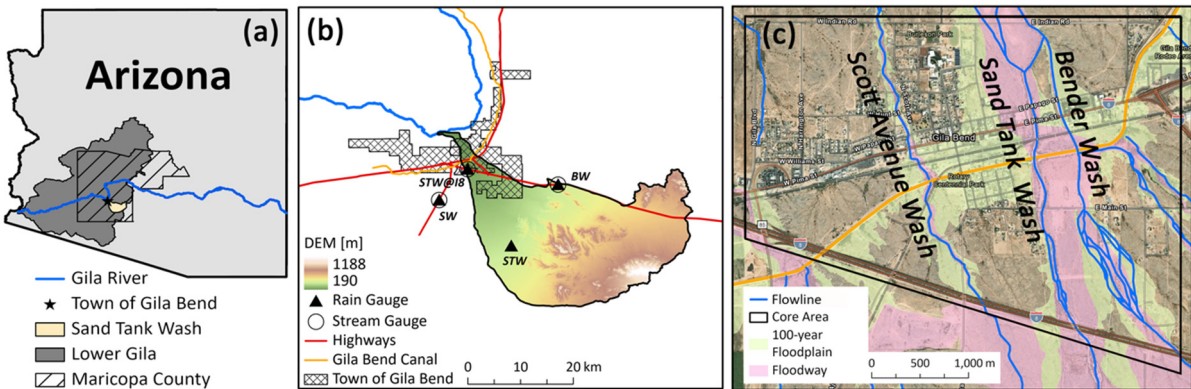

**Figure 1.** Study area. (**a**) Location of the town of Gila Bend in Arizona. (**b**) Digital elevation model (DEM) of the Sand Tank Wash watershed, with relative locations of key infrastructure, precipitation gauges, and streamflow gauges. (**c**) The core area of Gila Bend, overlaid with flowlines from the USGS and 100-year floodplain and floodway map from FEMA.

The town of Gila Bend has been subject to repeated flooding for decades (Table 1). Its urban areas are located downstream of several watersheds from which floodwaters can originate in the mountains to the southeast. The washes drain north-northwest to the Gila River. Among those 10-digit watersheds determined by the U.S. Geological Survey, the

Sand Tank Wash (STW) is the largest (681 km$^2$) with elevations ranging from 190 to 1188 m (Figure 1b). Potential flood hazards are exacerbated by infrastructure, including Interstate 8 (I-8), State Route 85 (SR85), the Union Pacific Railroad tracks, and the Gila Bend Canal (GBC), which have a significant impact on drainage flow patterns.

**Table 1.** Flood history events in the town of Gila Bend, Arizona.

| Date | Rainfall [mm/day] | Hazards | Record Types |
|---|---|---|---|
| 13–15 August 1990 | 30 | Ponding along GBC and SR85 | Photos |
| 27 August 2013 | 38 | SR238 closed due to flooding | Twitter |
| 4 May 2015 | 14 | SR238 closed due to flooding | Twitter |
| 13–14 September 2015 | 51 | Trapped 8–10 cars and up to 30 passengers | Twitter, photos |

Our analyses focused on the core area of Gila Bend (Figure 1c, 13 km$^2$ in area) which is centered on the most developed urban sites, bounded by I-8 to the south, Indian Road to the north, and Gila Boulevard to the west. The GBC flows from the northeast to the southwest corner of the core area. In the core area, 25% of the land is classified in the 100-year floodplain, with an additional 18% classified as floodways in the current FEMA (Federal Emergency Management Agency) map. Within the core area, there are three main floodways, namely Scott Avenue Wash, Sand Tank Wash, and Bender Wash. The Sand Tank Wash has the maximum drainage capacity of 318 m$^3$/s, followed by Bender Wash (104 m$^3$/s) and Scott Avenue Wash (83 m$^3$/s), at locations just downstream of I-8, according to the Flood Control District of Maricopa County (FCDMC).

*2.2. 13–14 August 2021 Flash Flood Event*

The 2021 monsoon season was one of the wettest on record in the arid Southwest U.S. [35], with several notable storm events in Arizona. Among them, a severe storm developed during the late evening hours of 13 August and continued into the morning of 14 August 2021. Initially developed over higher terrains, the storm swept across Maricopa County, creating strong, widespread winds, heavy rainfall, ponding, and flash flooding in the region. The town of Gila Bend was the worst affected area with torrential rainfall (around 100 mm in two hours) causing significant flooding. Peak discharge at one gauging station (STW@I8) was 396 m$^3$/s, which corresponds to a 60-year event [36].

Media reports indicated that 130 homes, 17 businesses, and the railway tracks were damaged by the storm [37]. More than 100 people were evacuated from their homes, with 30 people rescued from rooftops by emergency crews [38]. Two fatalities were also reported. A State of Emergency was declared by the Mayor of Gila Bend on the morning of 14 August, and a Declaration of Emergency was declared by the Governor of Arizona on 16 August, in anticipation of additional heavy rains in Maricopa County [39].

*2.3. Hydrometeorological Data*

We obtained observational datasets from precipitation and streamflow gauges operated by FCDMC (Table 2). Half-hour rainfall records were derived from a set of four precipitation gauges in the area, with their locations marked in Figure 1b. In addition to ground precipitation gauges, we obtained rainfall estimates from Multi-Radar Multi-Sensor (MRMS) precipitation product [40]. Derived by integrating radar and gauge observations, MRMS characterized the spatial distribution of rainfall at 4 km resolution.

**Table 2.** Precipitation (P) and streamflow (Q) gauges operated by FCDMC.

| Station Name | Abbr. | Dev. Type | Elevation (m) |
|---|---|---|---|
| Sand Tank Wash | STW | P | 354 |
| Sand Tank Wash @ I-8 | STW@I8 | P & Q | 232 |
| Sauceda Wash | SW | P & Q | 256 |
| Bender Wash | BW | P & Q | 366 |

### 2.4. PlanetScope Imagery

PlanetScope (PS) data are acquired by a constellation of CubeSats operated by Planet Labs [34]. All Planet satellites are in a sun-synchronous orbit at 475 km, with overpass time between 9:30 and 11:30 a.m. local time. The CubeSat fleet provides imagery in the visible (red, blue, and green) and near-infrared (NIR) bands with 3 m resolution on a near daily basis. With the combination of very high spatial and temporal resolution, PS imagery is emerging as a useful tool to monitor fast changes in surface conditions, such as vegetation phenology [41,42], river ice velocity [43], surface water areas [44,45], and streamflow presence [30] across different climatic zones. In arid regions, change detection of surface water is aided by sharp contrasts between wet and dry areas [30].

We used the radiometrically-, sensor-, and geometrically-corrected PlanetScope Analytic Ortho Tile (PSOrthoTile) product. This also generated a surface reflectance (SR) product from the top of atmosphere reflectance data using the 6SV2.1 radiative transfer model. We obtained PS imagery for one clear day before (5 August 2021) and for two days after the flood (15 and 16 August 2021). All images were captured by the newest generation of PS sensors (PSB.SD) that have interoperable spectral bands with Sentinel-2 [33,46]. We also obtained PS imagery for another storm event (2-year return period) on 19 September 2018 to compare with the flood extent of the 13–14 August 2021 storm.

### 2.5. Flood Extent Mapping

We mapped flood areas based on differences in NIR SR before and after the flood:

$$\Delta NIR = NIR_{Flood} - NIR_{Dry} \tag{1}$$

The key assumption of this method is that wet or ponded areas after the flood should have lower NIR SR (i.e., negative $\Delta NIR$), since water is a strong absorber of SR in the NIR band. In addition, when a clear PS image was acquired after the event (labeled $NIR_{Flood}$), many flooded areas had already dried out (i.e., no standing water), and surfaces were covered with sediments transported from upstream locations. For areas that had already dried when the CubeSat imagery was acquired, responses of $\Delta NIR$ depended on the relative magnitude of NIR SR between the original surface material and the new surface cover after the flood, including mud and debris transported from nearby terrains. To validate the derived flood extent, we obtained two-dimensional hydraulic simulations under different storm frequencies (2, 5, 10, 25, 50, and 100-year) using the FLO-2D model (https://flo-2d.com, accessed on 31 July 2022) that is approved by FEMA for the estimation of flood hazards within the jurisdiction of FCDMC.

## 3. Results

### 3.1. Hydrometeorological Event

Severe monsoon thunderstorms in central Arizona have common patterns characterized by large-scale synoptic conditions [18,47]. The 13–14 August 2021 storm was a Type-II event, identified by [47], which was associated with a high-pressure system over the Great Basin (Utah and Nevada) and an unusual upper-level trough over the eastern U.S. that positioned a cold front at the northern part of the New Mexico–Arizona border. On the evening of 13 August, scattered thunderstorms first developed over the Mogollon Rim and White Mountains and were then pushed into the lower desert area by the northeast winds generated from the clockwise circulation. Meanwhile, the extremely wet (e.g., total precipitable water of 24 to 50 mm) and unstable (convective available potential energy of 1500–2000 J/kg) atmosphere in southeastern to central Arizona favored the setup of severe storms capable of producing significant rainfall and flash floods [48].

In the late evening of 13 August, additional storm clusters developed across the Phoenix metropolitan area, producing high amounts of rainfall, water ponding, and flash flooding. During the event, the National Weather Service offices in Phoenix, Flagstaff, and Tucson issued a total of 29 flash flood/flood warnings, indicating the large spatial coverage

of the event. Clusters of storms developed into a Mesoscale Convective System (MCS) over south-central Arizona at midnight [49]. As the MCS slowly moved south, areas to the east and south of Gila Bend experienced torrential rainfall and significant flooding. Most of the rainfall fell around 1:00–2:00 a.m. local time, with maximum rainfall intensities reaching 50 mm/h (Figure 2a). Multiple FCDMC gauges received 38 to 102 mm of total rainfall (with the highest amount at BW). According to the FCDMC [36], the 1 h rainfall reached the 1000-year return period at the BW station. At the daily scale, the return period of rainfall was 306 years at BW, and 25 to 50 years at STW and SW.

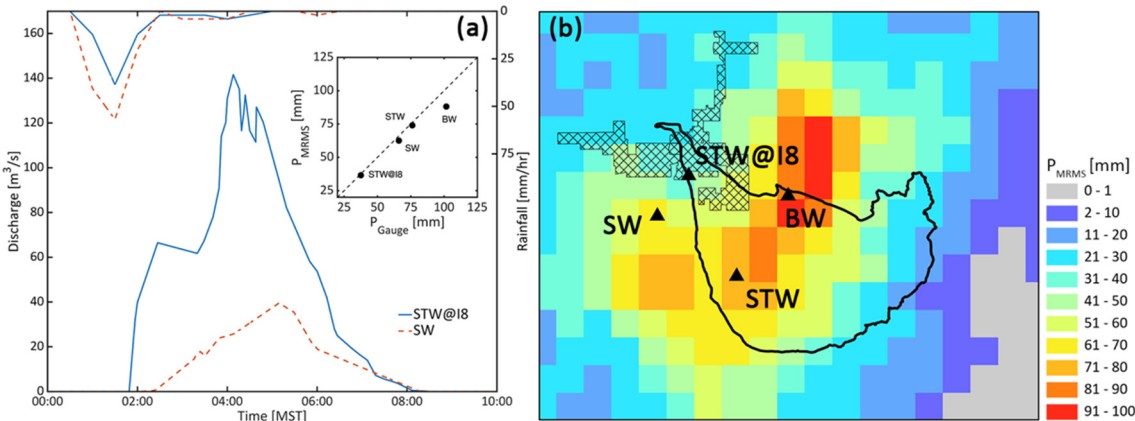

**Figure 2.** (**a**) Time series of rainfall (top axis) and streamflow (bottom axis) observations at STW@I8 (blue lines) and SW (red lines) stations, with the inset comparing the total precipitation during the storm between gauge records and the MRMS product. (**b**) Map of total precipitation from MRMS during the storm event, with the town of Gila Bend and Sand Tank Wash watershed shown for reference. The size of each MRMS pixel is 4 km.

Radar-derived rainfall (Figure 2b) was useful in describing the spatial distribution of precipitation accumulation during the storm in the vicinity of Gila Bend and its upstream watersheds. Clearly, the Sand Tank Wash received among the highest rainfall amounts (70 to 100 mm). The torrential rains led to a rapid surge of flooding (Figure 2a), which broke the record streamflow at the STW@I8 station.

### 3.2. Flood Mapping from CubeSat RGB Imagery

The flood wave generated south of Gila Bend traveled along three main floodways, namely Scott Avenue Wash, Sand Tank Wash, and Bender Wash (from west to east). All three washes intersect highway I-8. True color PS imagery on 15 August (one day after the flood, Figure 3b) showed that most of the floodways downstream of I-8 were occupied by the flow. In addition, several cross sections along I-8 had discharges that likely exceeded the infrastructure drainage capacity, thus creating large, ponded areas upstream of the highway. In Gila Bend, flood damage was most severe in areas south of Pima Street. Visual evidence of ponding was noted on both sides of the GBC as the conveyance capacities of culverts across the canal were not adequate to transport the 60-year event. Additional ponded areas were also observed to the east of the GBC. The BW wash did not record a very high discharge (11 m$^3$/s) but might have exacerbated flooding along the STW. From Gila Bend, flooding continued northward to enter the Gila River.

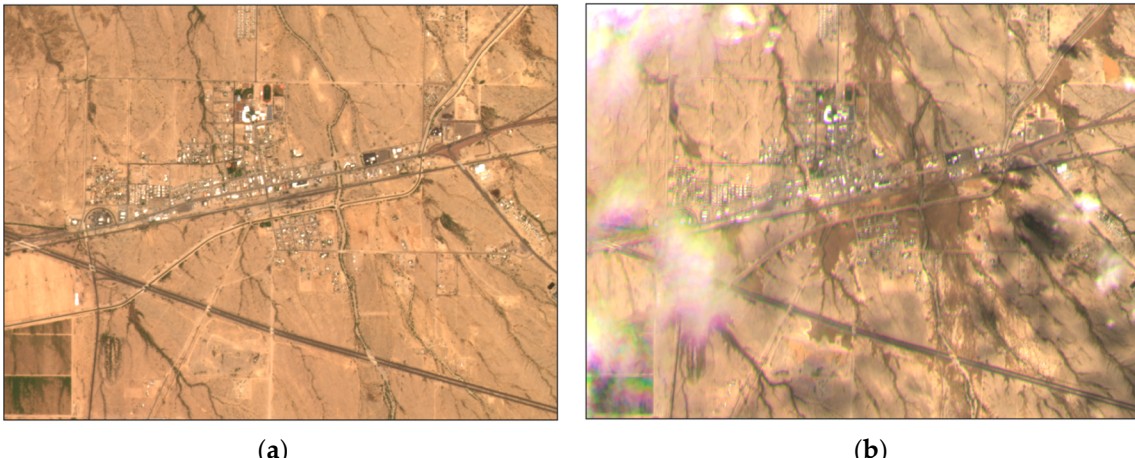

<center>(<b>a</b>)              (<b>b</b>)</center>

**Figure 3.** Satellite aerial imagery of Gila Bend from (**a**) 8 May 2021 (dry condition), and (**b**) 15 August 2021 (1 day after the flood event), captured by PS imagery.

### 3.3. Flood Mapping from CubeSat NIR Imagery

In addition to visually examining the flood extent, we applied the flood mapping method using the NIR SR data. We focused on the core area of Gila Bend, where we compiled existing infrastructure drainage capacities at key locations (channels and crossings). In addition to the 13–14 August 2021 event, we first analyzed a 2-year flood event on 19 September 2018, with a peak discharge of 27.5 m$^3$/s at the STW@I8 bridge site.

Figure 4 shows the changes in NIR SR ($\Delta$NIR) before and after the events for the 2018 and 2021 cases. As expected, changes in $\Delta$NIR were more significant over areas with ponded water. For the 2018 event, areas inside channels showed a much lower $\Delta$NIR than those outside the channel (Figure 4b). The contrasting behavior of $\Delta$NIR when water was present in the channel was also observed in a nearby ephemeral river [30]. By using an $\Delta$NIR threshold of $-0.05$, a flood extent map of the 2018 event was generated (Figure 4e). Total flooded areas (i.e., $\Delta$NIR $\leq -0.05$) accounted for 0.9% of the core area, with most located inside the floodway and the 100-year floodplain (94% and 3%, respectively), which is expected as the peak runoff was well below the drainage capacity of the channel.

In contrast, for the 2021 flood event, $\Delta$NIR displayed a more complicated spatial pattern (Figure 4c). Widespread areas showed a negative $\Delta$NIR, indicating ponding which far exceeded the flooding in the previous case. In addition, other locations within the core area had a positive $\Delta$NIR after the flood. This occurred for multiple reasons. First, flood waters damaged parts of the GBC and the local manager shut off the water to the canal, essentially reducing canal flows after the storm. Second, many town roads had positive changes in $\Delta$NIR. This was likely due to the urban surfaces prior to the event having lower SR than after the flood debris covered them. At the overpass time, flood debris had not been cleared from roads. Third, some areas outside the town were covered with transported sediments that had already dried by the overpass time (~50 h after the runoff ended). To account for these different factors, we classified areas with $\Delta$NIR $\leq -0.05$ or $\Delta$NIR $\geq 0.02$ both as "flooded", with the former considered as Type I (ponded water) and the latter as Type II (flood impacted due to debris) areas (Figure 4d).

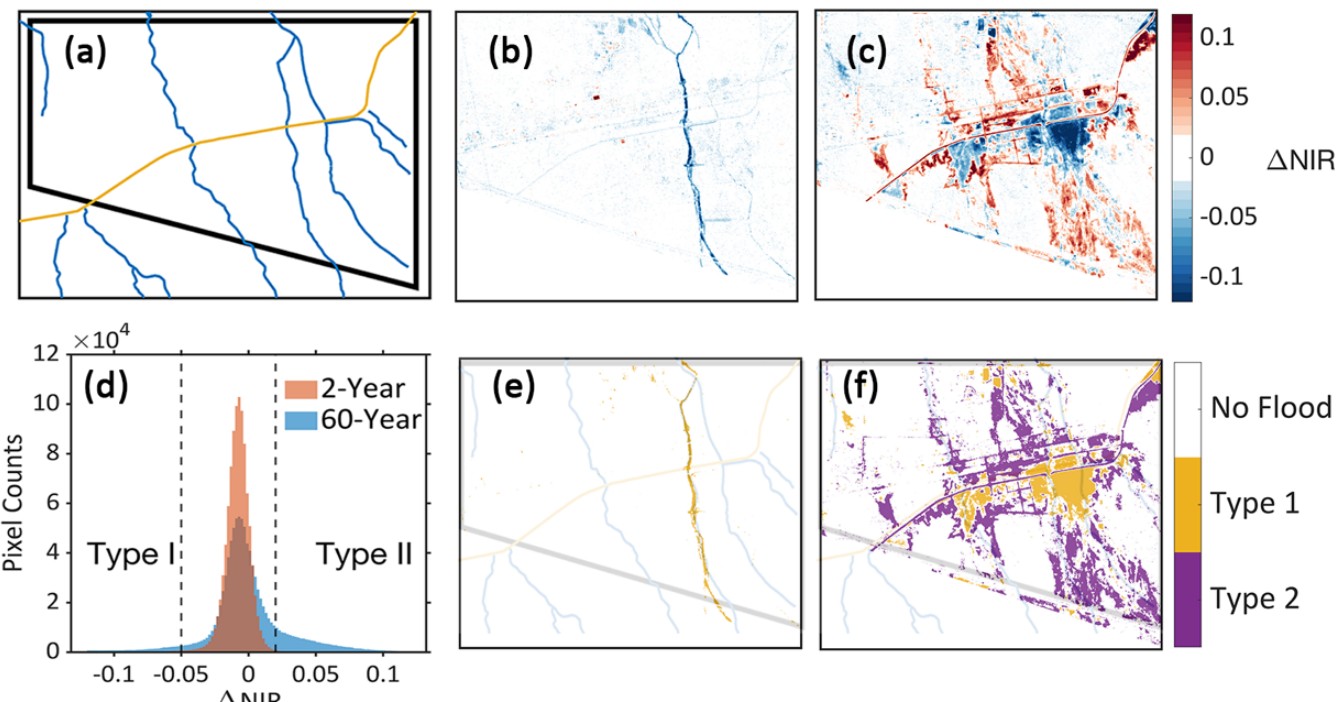

**Figure 4.** Flood-affected area from CubeSat imagery. (**a**) The boundary of the core area, with blue flowlines and the yellow Gila Bend Canal. (**b**) ΔNIR for the 2018 event (2-year return period). (**c**) ΔNIR for the 2021 storm event (60-year return period). (**d**) Histogram of ΔNIR for two events. (**e**) Flooded area in 2018 event, with (**a**) as reference in background. (**f**) Same as (**e**) but for 2021 event. Type I ponded areas have ΔNIR ≤ −0.05 whereas Type II flood-impacted areas have ΔNIR ≥ 0.02.

FCDMC gauging records showed the peak flow rate at STW@I8 was roughly 396 $m^3$/s, close to the 100-year discharge estimate (409 $m^3$/s). According to the FCDMC field inspection records, the wide flow "may have consumed much of the floodway" downstream of I-8 [36]. This is confirmed with the Type II flooded areas between I-8 and Main Street, as mapped by CubeSat imagery (Figure 4f). Further downstream, there were many Type I flooded areas on both sides of STW. Due to the limited conveyance capacity of the overchute at the canal (204 $m^3$/s), surface runoff overflowed from the main STW channel and caused severe overbank flooding in the core area. Moreover, the higher elevation of both the canal embankment and the railway created a series of dam structures that turned surrounding areas into multiple ponds. Neighborhoods in this region, especially those located south of the canal, have poor drainage that caused extensive local inundation mapped as Type II flooded areas where flood debris covered urban surfaces.

To evaluate the accuracy of the CubeSat-derived flood extent map, we obtained the simulated flow depths from a two-dimensional hydraulic model (FLO-2D, [50]) applied to the region as part of the Gila Bend Area Drainage Master Plan. Modeling results consider existing infrastructure in the area and its interaction with the 100-year flood, which had similar discharge values at the STW@I8 site to the 13–14 August 2021 event. We qualitatively compared the flood extents from the FLO-2D simulations and CubeSat mapping. Flooded areas from the simulation (flow depth > 10 cm, Figure 5a) matched well with those derived from CubeSat imagery (Type I and II, Figure 5b), demonstrating the capability of the threshold-based ΔNIR approach to map flood extents.

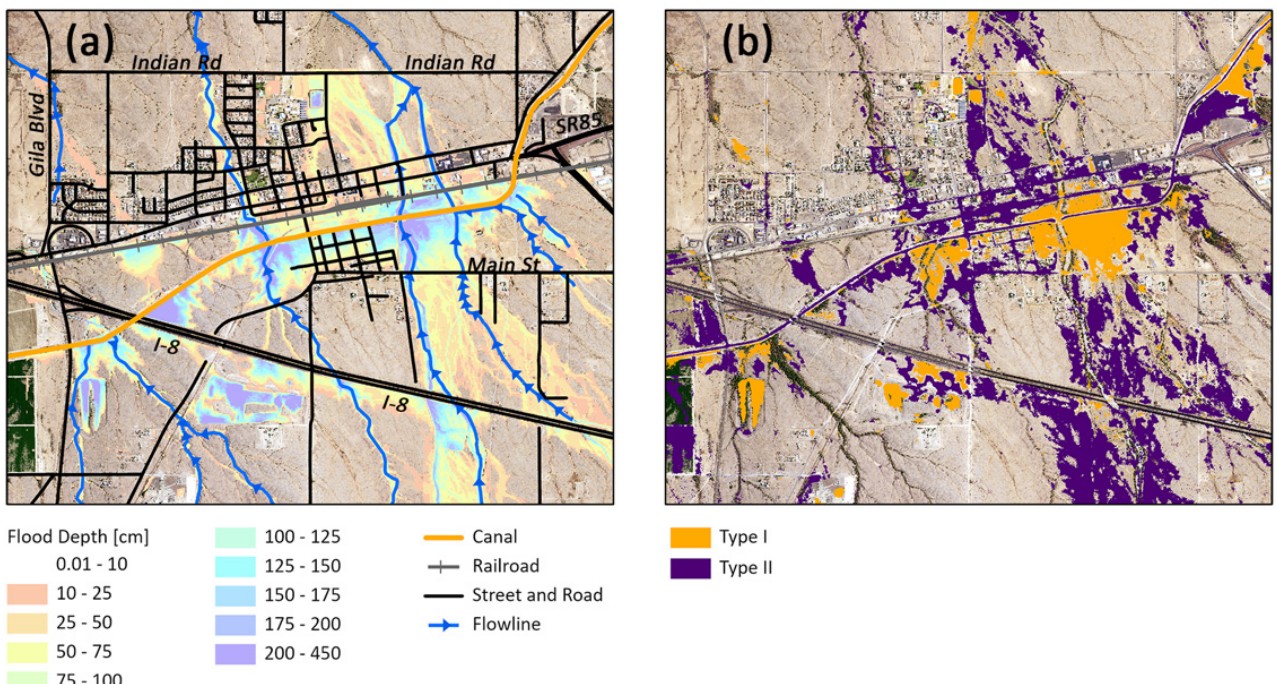

**Figure 5.** Flood extents around the town of Gila Bend from different sources. (**a**) Flood depth under 100-year return period simulated by FLO-2D, with streets, roads, railroad, canal, and flowlines labeled for reference. (**b**) Flood extent for two types of flooded areas derived from CubeSat imagery.

As mentioned previously, the two key features of CubeSat flood detection in arid regions are high spatiotemporal resolution and sharp contrasts between flood-impacted areas and drier surrounding zones that allow identifying robust changes in NIR before and after a flood. Figure 5 shows that areas with higher flooding depths (>1 m) were located near the major intersections of the different washes with the GBC and with highway I-8, which both serve as barriers to streamflow. As a result of higher flow depth, it is likely that ponded water in these area lasted a relatively long time. This can explain the good match between the spatial distribution of high-flow depths from FLO-2D and the Type I flooded areas identified from CubeSat imagery. In comparison, areas with lower flow depths were generally classified as Type II flooded areas as these were primarily impacted by flood debris and had dried out by the time of the CubeSat overpass.

## 4. Discussion

### 4.1. Hydrometeorological Hazard Detection Using CubeSats

Remote sensing imagery has been used in hydrometeorological hazard detection for almost half a century since the launch of LandSat-1 [28]. However, it has not been until recently that gaps in spatiotemporal measurements have been filled by observations from CubeSats, for instance from Planet Labs [51]. In this study, the high temporal resolution of CubeSat imagery allowed a timely mapping of the flood-affected areas when spectral changes in surface conditions were still detectable from space. This method relied on the near-infrared band that was able to identify changes in surface properties due to ponded water, wet soils, and debris on the surfaces of infrastructure and natural areas. The high spatial resolution from CubeSats also afforded the capability to map the flood extent in detail and relate the spatial patterns to natural channels and floodways, transportation and water resources infrastructure, and the predicted flood depths from a 100-year storm event obtained using a two-dimensional hydraulic model application. In contrast to these capabilities, traditional satellites provide much longer revisiting periods and at typically coarser spatial resolutions, such that the impacts of flash flood events can be missed. The analysis performed here revealed the unique advantage of CubeSat imagery for rapidly

mapping the effects of hydrometeorological hazards. Furthermore, the CubeSat detection approach is likely to perform well year-round in arid regions where cloud-free optical remote sensing images have a high likelihood to be acquired before and after flash flood events. While this study was performed for central Arizona, the study approach can be readily applied in other regions as drylands constitute ~40% of the Earth's surface [50].

### 4.2. Potential Applications for Post-Flood Analyses in Arid Regions

Mapping flash flood hazards in arid regions from CubeSats can lead to an improved understanding of hydrological processes as well as support for water resources decision-making. For instance, the performance of two-dimensional floodplain models can be evaluated through comparisons to the flood-affected areas determined from CubeSat data for different storm events [52–55], as shown here. Compared with traditional validation approaches that rely on in situ measurements, the flooding extent offered by CubeSat remote sensing can build confidence in the spatial model performance and corroborate other types of observations obtained from affected citizens, news media reports or post-flood surveys [26,56–58]. In addition, most reported damaging flood events are now located outside FEMA-delineated high-risk zones in unmapped areas of the United States [59]. Under these circumstances and within the constraints and limitations of the methodology, as discussed next, CubeSat-derived flood extent maps can identify flooded areas after major events, help determine flood insurance rates, and supplement information in floodplains where other ground-based information is not readily available [60].

### 4.3. Limitations and Uncertainties

There are several important limitations of CubeSat imagery that limit its application for flash flood hazard detection. Despite the near-daily revisiting frequency, cloud cover is still an intrinsic challenge when using optical imagery [61]. After certain flood events, cloud cover can obscure land surfaces completely for a long period. Scattered clouds can also introduce uncertainties in the analysis as cloud shadows are commonly misclassified as water [62]. Another limitation is that the historical archive of PlanetScope imagery only dates back to 2016, such that analyses of previous flooding events are not possible and a climatological record of flooding events based on CubeSat imagery is unavailable at the moment. Despite the above, quality control procedures that account for measurement uncertainties can overcome some limitations to enable robust flash flood hazard detection [30,44]. In addition, CubeSat products are constantly improving, including the use of new sensors that provide higher-quality imagery with more spectral bands [33]. As the archive of CubeSat imagery grows, the value of the historical record will improve and additional analyses will become possible, including comparisons across different events.

### 5. Conclusions

Our analysis of a flash flood event in the town of Gila Bend, Arizona, presents a first-of-its-kind mapping of flood extent in an arid region using CubeSat remote sensing. We described the generation and propagation of the storm event by integrating hydrometeorological data from multiple sources and then showed the spatial flood extent by comparing surface reflectance data before and after the flood. This was possible due to the unprecedented spatiotemporal resolution of true color RGB images and NIR surface reflectance from PlanetScope. Our analyses suggest the feasibility of CubeSat observations in capturing the flash flood extents in arid and semiarid regions where dramatic changes in surface reflectance occur between flooded areas and surrounding regions. As a result, this study fills an important gap in the reconstruction of extreme hydrometeorological events and can provide valuable information for testing hydrologic and hydraulic models and improving the understanding of flood responses to extreme storms.

**Author Contributions:** Conceptualization, Z.W. and E.R.V.; methodology, Z.W. and E.R.V.; formal analysis, Z.W. and E.R.V.; investigation, Z.W. and E.R.V.; data curation, Z.W.; writing—original draft preparation, Z.W. and E.R.V.; writing—review and editing, Z.W. and E.R.V.; visualization, Z.W. and E.R.V.; supervision, E.R.V.; project administration, E.R.V.; funding acquisition, E.R.V. All authors have read and agreed to the published version of the manuscript.

**Funding:** This research was funded by NASA Commercial Satellite Program (Determining Streamflow Regimes from Commercial Smallsat Data in Arid and Semiarid Regions, 80NSSC21K1154).

**Data Availability Statement:** PlanetScope imagery is available for scientific purposes from the Smallsat Data Explore Application (https://csdap.earthdata.nasa.gov, accessed on 15 July 2022). Precipitation and streamflow data can be obtained from the Flood Control District of Maricopa County (https://www.maricopa.gov/5308/Flood-Control-District, accessed on 15 July 2022).

**Acknowledgments:** PlanetScope imagery was made available through the Planet Incubator Program at Arizona State University and the Smallsat Data Explore Application (https://csdap.earthdata.nasa.gov, accessed on 15 July 2022), part of the NASA Commercial Smallsat Data Acquisition Program. We appreciate the excellent storm reports prepared by the National Weather Service (Phoenix Office) and the Flood Control District of Maricopa County (Engineering Division, Flood Warning Branch) as well as the FLO-2D modeling results provided by FCDMC.

**Conflicts of Interest:** The authors declare no conflict of interest.

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
