# Peer review of "Mapping Flash Flood Hazards in Arid Regions Using CubeSats"

_remotesensing, doi:10.3390/rs14174218_

Round 1
Reviewer 1 Report
Dear Authors,
The authors presented an approach of “Mapping Flash Flood Hazards in Arid Regions using CubeSats.” The subject is interesting and within the scope of the journal: Remote sensing. However, I think significant concerns are reasonable and could improve the manuscript’s quality before publication. Overall, I would recommend publication of this manuscript subject to “minor revision” −taking into account precisely addressing the following comments in the revised version.
1. I’d suggest adding model/result validation in the abstract.
2. I am not sure what is the main research hypothesis.
3. Regarding the model/method, I would be happy to see more details regarding splitting the data. Train, Validation, and Testing subsets should be better explained and justified in the context of this workflow. I could not find anything related to model validation. A separate paragraph can be introduced stressing the validation section.
4. The contents sometimes jump between topics without a clear direction in the discussion section. I would have wished to see more information on the actual meaning of the findings and how the results add to the broader issue and the specific scientific field.
Author Response
Our responses to the reviewer comments are as follows, based on the numbering scheme cited below.
- We have improved the abstract by adding information on the independent study from which we obtained the 100 year flood depths. This change includes: "The unprecedented spatiotemporal resolution of CubeSat imagery allowed the detection of ponded (ΔNIR ≤ -0.05) and flood-affected (ΔNIR ≥ +0.02) areas that compared remarkably well with the 100-year flood event obtained through an independent hydraulic modeling study."
- The main hypothesis is that CubeSats can be used to map flooding hazards in arid regions. We have now included this more clearly in the abstract as: "In recent years, the CubeSat constellation operated by Planet Labs fills this key gap in Earth observation by providing 3-m, near-daily multispectral imagery at the global scale. In this study, we demonstrate the imaging capabilities of CubeSats for mapping flash flood hazards in arid regions."
- In this study, we did not train the hydraulic model or directly derived flooding areas from a modeling approach. As a result, this study did not involve data splitting into calibration and validation periods. Our approach is entirely observational and remote sensing based. An independent estimate of the 100 year flood conditions for the study site was obtained from the local flood authority. We now describe this in the methods as: "To validate the derived flood extent, we obtained two-dimensional hydraulic simulations under different storm frequencies (2, 5, 10, 25, 50, and 100-year) using FLO-2D model that is approved by FEMA for mapping of flood hazards within the jurisdiction of FCDMC."
- We have added transitions in the discussion section to guide the readers. For instance, we added: "While this study is performed for central Arizona, the study approach is readily applied in other regions since drylands constitute ~40% of the Earth’s surface.". We also modified: "Under these circumstances and within the constraints and limitations of the methodology, as discussed next,". Subsequently, we added a new section that outlines the major limitations and uncertainties of the methodology.
Reviewer 2 Report
This study is well-designed. This paper is well-written and well-referenced. However, the following minor corrections may further enhance the quality of this manuscript:
1. One keyword (flood hazard) duplicates the same as in the paper title. Another selection should be re-chosen.
2. Since this study is performed in very specific conditions (study area, limited cases), the limitations of this study success should be emphasized.
3. To enhance the contribution of this study, a paragraph on recommendation for further extension and generalization may be helpful.
Author Response
Our responses to the reviewer comments are as follows, based on the numbering scheme cited below.
- We removed the keyword 'flood hazard' and replaced with 'hydrometeorology'.
- We added a new subsection in the discussion (4.3 Limitations and Uncertainties) to present the limitations and uncertainties of the approach. We also added a sentence that directly addressed the concern about the limited study area or case study as: "While this study is performed for central Arizona, the study approach is readily applied in other regions since drylands constitute ~40% of the Earth's surface."
- We discussed potential applications in Section 4.2 which we deem is sufficient to identify useful future avenues of research.
Reviewer 3 Report
The paper is generally well written and provides useful insights for drylands water managers and ecohydrologists. My major concern is about the abstract and adding some implications of the findings at the end of the abstract and discussion parts. So this leads me to recommend accepting after minor revisions. I hope my review can help to improve the quality of the paper. Please find below some specific comments.
- Abstract is too short and needs improvement according to its extent.
- Lines 49-52: Add citations here.
- Lines 55-60: Delete these sentences.
- Line 64: Use SI metric (i.e., km).
- Please add the implication/limitation of such a study and briefly indicate the advantage or disadvantage.
- Conclusion: The conclusion will be necessary to have bigger details.
Author Response
Our responses to the reviewer comments are as follows, based on the numbering scheme cited below.
- We added references 32, 33 and 34 to this sentence.
- We believe these sentences are important for the reader to understand the structure of the paper and have retained them.
- We changed the unit to km.
- We had provided a robust discussion of the implications and the potential for the use of CubeSats to monitor flash flooding. We have now added a subsection to the discussion (Section 4.3) to outline the limitations and uncertainties of the method.
- Since a conclusions section is not mandatory for MDPI journals, we believe a short concluding section as provided here is sufficient.